# Neuroimaging Study of Brain Functional Differences in Generalized Anxiety Disorder and Depressive Disorder

**DOI:** 10.3390/brainsci13091282

**Published:** 2023-09-04

**Authors:** Xuchen Qi, Wanxiu Xu, Gang Li

**Affiliations:** 1Department of Neurosurgery, Sir Run Run Shaw Hospital, Zhejiang University School of Medicine, Hangzhou 310000, China; qixuchen@zju.edu.cn; 2Department of Neurosurgery, Shaoxing People’s Hospital, Shaoxing 312000, China; 3College of Engineering, Zhejiang Normal University, Jinhua 321004, China; xwx@zjnu.edu.cn; 4College of Mathematical Medicine, Zhejiang Normal University, Jinhua 321004, China

**Keywords:** generalized anxiety disorder (GAD), depressive disorder (DD), electroencephalogram (EEG), beta rhythm, psychiatry, neuroimaging, functional brain imaging, neurosciences, machine learning, diagnosis

## Abstract

Generalized anxiety disorder (GAD) and depressive disorder (DD) are distinct mental disorders, which are characterized by complex and unique neuroelectrophysiological mechanisms in psychiatric neurosciences. The understanding of the brain functional differences between GAD and DD is crucial for the accurate diagnosis and clinical efficacy evaluation. The aim of this study was to reveal the differences in functional brain imaging between GAD and DD based on multidimensional electroencephalogram (EEG) characteristics. To this end, 10 min resting-state EEG signals were recorded from 38 GAD and 34 DD individuals. Multidimensional EEG features were subsequently extracted, which include power spectrum density (PSD), fuzzy entropy (FE), and phase lag index (PLI). Then, a direct statistical analysis (i.e., ANOVA) and three ensemble learning models (i.e., Random Forest (RF), Light Gradient Boosting Machine (LightGBM), eXtreme Gradient Boosting (XGBoost)) were used on these EEG features for the differential recognitions. Our results showed that DD has significantly higher PSD values in the alpha1 and beta band, and a higher FE in the beta band, in comparison with GAD, along with the aberrant functional connections in all four bands between GAD and DD. Moreover, machine learning analysis further revealed that the distinct features predominantly occurred in the beta band and functional connections. Here, we show that DD has higher power and more complex brain activity patterns in the beta band and reorganized brain functional network structures in all bands compared to GAD. In sum, these findings move towards the practical identification of brain functional differences between GAD and DD.

## 1. Introduction

Mental disorders, such as Generalized Anxiety Disorder (GAD) and Depressive Disorder (DD), cast a profound shadow on individuals’ lives, influencing their overall well-being and leaving a lasting impact. Conditions like GAD and DD can significantly disrupt an individual’s emotional equilibrium, cognitive functioning, and overall quality of life [1]. GAD is characterized by excessive and persistent worry or anxiety about various aspects of life, including work, relationships, health, and everyday situations [2,3,4]. Based on previous reports, within urban areas of China, the prevalence of GAD is reported to be 5.3% [5], highlighting its substantial presence in the population. DD involves persistent feelings of sadness, hopelessness, and a loss of interest or pleasure in activities [6]. There are approximately 150 million people in the world who are suffering from DD at present, underscoring the global scope of this mental health challenge [7]. While GAD and DD share some similar symptoms [8], they are distinct conditions with unique underlying neurobiological mechanisms [9]. The comprehension of specific functional brain differences in imaging between GAD and DD assumes a critical role in achieving accurate diagnoses, tailoring personalized treatment strategies, and ultimately enhancing the well-being and outcomes for affected individuals.

Traditionally, the diagnosis and differentiation of GAD and DD has heavily relied on clinical assessments, subjective reports, and symptom-based criteria. However, advancements in machine learning techniques and neuroimaging technologies have opened up new avenues for exploring objective biomarkers and identifying the brain functional differences associated with these disorders [10,11,12,13]. Machine learning algorithms have demonstrated potential in analyzing neuroimaging data to identify patterns and features that can be used to differentiate between GAD and DD [14,15,16]. Algorithms can extract intricate information from neuroimaging modalities such as electroencephalography (EEG) [17], enabling a deeper understanding of the unique brain signatures associated with each disorder [18,19,20]. By leveraging these advancements, the field of mental health research is making strides toward objective and quantifiable measures. These measures can potentially transcend the confines of subjective reporting and allow for more accurate, reliable, and early detection. As machine learning algorithms continue to be refined and neuroimaging technologies progress, the boundaries of what can be discerned from brain activity patterns expand, fostering a new era of precision psychiatry. In this emerging landscape, the amalgamation of sophisticated analysis and cutting-edge technology stands to enhance our understanding of mental disorders like GAD and DD, offering the potential for more targeted interventions and improved outcomes for those affected.

EEG is a non-invasive procedure with the advantages of lower costs, higher temporal resolution and more convenient operation, which helps in the diagnosis and monitoring of diverse neurological conditions and disorders. Its recordings provide valuable information about brainwave patterns, allowing for healthcare professionals to identify abnormalities and make diagnostic interpretations [21,22,23]. Given the above strengths, EEG has been extensively utilized for detecting and diagnosing mental disorders, such as DD [24,25], and GAD [23,26]. Power spectrum density (PSD) analysis is one of the primary methods of EEG, which can calculate several features (power, relative power, power ratio, etc.) [27]. Specifically, Oathes and his team found that, in the posterior channel, high power was shown in the gamma band in GAD patients [14]. This observation suggests distinctive neural activity patterns in individuals grappling with GAD. Dell reported that the power in the delta band may correlate with the hypoactivation of the motivation system, related to the proximity in the patients with DD patients [28]. These insights illuminate potential neurobiological markers that may contribute to the manifestation of these disorders. In addition, fuzzy entropy (FE) is a measure that characterizes the complexity of brain networks and provides valuable insights into the dynamic information of the brain. One advantage of FE is that it does not depend on the length of the time series, making it particularly relevant for practical applications [23]. A higher value of fuzzy entropy indicates a higher level of complexity. Previous studies have found that DD patients show lower complexity than the healthy control group [29]. This implies that the intricate interactions within brain networks may be perturbed in individuals with DD, underscoring the potential utility of FE as a diagnostic tool. Functional connectivity is a feature utilized in the analysis of brain connectivity that is derived from EEG signals [30,31]. Although more attention has been paid to the interactions of different brain areas, the relationships between brain areas in GAD and DD patients remain unclear [32]. To address this issue, Phase Lag Index (PLI) is an optimal analytical approach that can reveal the relationships and interactions within brain regions. The PLI quantifies the asymmetry of the phase differences between different EEG channels, allowing for insights into the functional connectivity and synchronization between brain regions [11,33,34]. Specifically, the focus of the PLI is on the phase information contained within the EEG signals, as this is believed to reflect the underlying neural communication and coordination.

The potential to revolutionize the field of psychiatric diagnostics lies in the integration of machine learning techniques with neuroimaging data [35,36]. Complex patterns and subtle interactions within brain networks can be learned by machine learning algorithms [37], leading to the objective and accurate identification of brain functional differences between GAD and DD. A growing body of the literature is reporting on the use of machine learning algorithms combined with EEG for the accurate diagnosis of GAD or DD [23,38,39]. Based on precise machine learning models, we can more accurately identify features with significant differences between GAD and DD from multidimensional EEG features. With the help of these advancements, earlier and more precise diagnoses can be facilitated, and tailored treatment strategies can be formed for individuals with GAD or DD.

The objective of this study was to develop an enhanced screening framework that can effectively differentiate between GAD and DD. To achieve this, multidimensional electroencephalography (EEG) features were utilized in the analysis, which involved integrating the evaluation of functional connections among different brain regions with conventional measures such as PSD and FE. By combining these perspectives, a comprehensive analysis framework that incorporated multiple EEG features was established and analyzed via statistical analysis and machine learning. Given the limited previous research focused on GAD and DD using this approach, the aim was to propose a refined framework and identify potential biomarkers that could contribute to accurate screening and diagnosis.

## 2. Materials and Methods

### 2.1. Subjects

Thirty-eight patients with GAD and thirty-four patients with DD, aged between 18 and 55 years old, were enrolled from the psychiatry department of the hospital. They were diagnosed by psychiatrists using the Structural Clinical Interview based on Diagnostic and Statistical Manual of Mental Disorder, Fifth Edition, (DSM-5) criteria. Each patient completed a questionnaire consisting of the Hamilton Anxiety Rating Scale (HAMA) for GAD and the Hamilton Depression Rating Scale (HAMD-17) for DD. As shown in Table 1, the scores were 17 points or higher. The HAMA score was 26.2 ± 5.8, and the HAMD-17 score was 26.3 ± 4.9. The average age for patients with GAD was 41.2 ± 7.2 (10 males and 28 females), while for patients with DD it was 42.3 ± 9.6 (10 males and 24 females). There was no statistically significant difference in age between patients with GAD and DD. Additionally, all patients were right-handed and had no other mental or physical disorders, as well as no history of substance or alcohol abuse. All participants were instructed to refrain from staying up late and consuming alcohol or drugs within one day prior to the test. No smoking, no coffee, and no tea were required at least eight hours before data recording.

### 2.2. Electroencephalogram Recordings

Ten consecutive minutes of EEG data were recorded by the EEG equipment at the specialized EEG labs in the Hospitals. Based on the 10–20 system, sixteen EEG channels, including FP1, FP2, F3, F4, C3, C4, P3, P4, O1, O2, F7, F8, T3, T4, T5 and T6, were placed on the scalp surface of the patient. EEG parameters were set as follows: sample frequency 250 Hz, band-pass filtering 0.5–30 Hz, time constant 0.3 s. During EEG recordings, participants were requested to keep relaxed, close the eyes, and concentrate their attention on breathing.

### 2.3. Electroencephalogram Data Preprocessing

EEG data preprocessing was conducted as follows. Firstly, remove EEG artifacts by fast-ICA, such as eye blinks, electrocardiograph, and electromyography. Secondly, lower the sampling frequency from 250 Hz to 125 Hz. Thirdly, remove baseline and implement bandpass filtering with 4–30 Hz by fourth-order Butterworth bandpass filter. Subsequently, the EEG data were segmented into 4 s samples with a 2 s overlap, yielding 8918 samples for GAD and 7121 samples for DD. Finally, EEG rhythms were extracted, including theta (4–8 Hz), alpha1 (8–10 Hz), alpha2 (10–13 Hz), and beta (13–30 Hz) for every sample with the same bandpass filter.

### 2.4. Multidimensional Electroencephalogram Features Computation

Numerous studies have documented the viability of employing EEG features for the identification of mental diseases [23,35,40]. To our knowledge, there are three mainstream analytical approaches to EEG feature extractions [41,42]: power spectral density analysis, nonlinear dynamics analysis, and functional connectivity analysis. These three approaches decode the information imbedded in EEG data from three different perspectives [23]. In this paper, an analytical framework for brain functional differences’ detection of GAD and DD using multidimensional EEG features was developed. For this purpose, three types of prevalent EEG features, including PSD, FE and PLI (which are the three most representative types of characteristics), are extracted in this study. The details are as follows.

#### 2.4.1. Power Spectrum Density Calculation

As shown in Equation (1), *x*(*i*) refers to the EEG signal, *X*(*f*) is the frequency spectrum of *x*(*i*) which can be estimated by FFT, and *P_x_*(*f*) is the power spectrum. As shown in Equation (2), *PSD*(*h*) is the EEG power for each rhythm. In Equations (1) and (2), *N* is the number point of *x*(*i*), *h* represents EEG rhythm, and *f_h_* and *f_l_* are the upper frequency and lower frequency, respectively, for each EEG rhythm.
(1)$$P_{x}(f)=\frac{1}{N}{\left|X(f)\right|}^{2}$$
(2)$$\mathrm{PSD}\left(h\right)=\frac{1}{f_{h}-f_{l}}\int_{f_{l}}^{f_{h}}Px\left(f\right)df$$

#### 2.4.2. Fuzzy Entropy Calculation

FE is a nonlinear EEG feature, which is used to measure the complexity and randomness of the EEG signals. For the computation of FE, firstly, the EEG signal is divided into an m-dimensional (m is the embedding dimension) subseries of length *N*, each of which is of the same length. Then the similarity $C_{i}^{m}$ is calculated between the subseries, as shown in Formula (3), where $d_{ij}$ denotes the fuzzy similarity between *i* and *j* subsequences, and *r* is a predefined parameter used to control the similarity. Finally, we can calculate FE for a piece of EEG sample through Formulas (4) and (5).
(3)$$C_{i}^{m}\left(r\right)=\frac{1}{N-m}\sum_{j=1,j\neq i}^{N-m+1}exp\left(-ln\left(2\right)\cdot {\left(\frac{d_{ij}}{r}\right)}^{2}\right)$$
(4)$$\emptyset^{m}\left(r\right)=\frac{1}{N-m+1}\sum_{i=1}^{N-m+1}C_{i}^{m}\left(r\right)$$
(5)$$FE\left(m,r,N\right)=ln\emptyset^{m}\left(r\right)-ln\emptyset^{m+1}\left(r\right)$$

#### 2.4.3. Phase Lag Index Calculation

Firstly, as shown in Formula (6), EEG signal *x_i_*(*t*) is transferred into *s_i_*(*t*) by Hilbert transform [43], where *S_i_* is the instantaneous amplitude and *φ_i_* refers to the instantaneous phase.
(6)$$s_{i}(t)=S_{i}(t)e^{j\varphi_{i}(t)}$$

Then, the PLI value of all pairs of EEG signals can be evaluated by Formula (7) [44]. As defined in Formula (7), PLI value ranges from 0 to 1. PLI = 0 means completely out of phase synchronization. Larger PLI values imply greater phase synchronization
(7)$$PLI_{k,l}=\left|\left\langle sign(\varphi_{k}(t)-\varphi_{l}(t))\right\rangle \right|$$

### 2.5. Ensemble Learning for Classification

Ensemble learning is a method of combining multiple weak machine learners to build a single strong machine learner, which follows different ideas, such as bagging and boosting. Bagging randomly selects a training subset on the given samples with equal weights on each sample, then independently trains the classification of different weak classifiers on each training subset, and finally obtains the final result by majority rule. Meanwhile, boosting adds larger sample weights to the samples that were misclassified in the previous weak classifier, and the final result is obtained by summing the results of multiple models. In current study, three ensemble learning models, Random Forest (RF), Light Gradient Boosting Machine (LightGBM), and eXtreme Gradient Boosting (XGBoost), are used for GAD and DD detection to identify significantly different EEG features and rhythms. These models are presented below.

(1) RF is a widely used ensemble learning model based on decision tree. RF uses the bagging ensemble strategy and random subspace method. RF includes many decision trees and integrates the results of each decision tree by averaging the votes in classification task or outputs in regression task to improve the stability and precision. A random perturbation of samples and attributes for each decision tree is achieved by introducing the bagging ensemble strategy and random subspace approach during the learning process. By randomizing the decision trees in these two dimensions, RF can solve the inherent overfitting problem of decision trees and obtain stable and reliable classification results.

(2) LightGBM is based on boosting ensemble strategy. LightGBM is an efficient machine learning algorithm that achieves good speedups on large datasets with no loss of accuracy. LightGBM uses a gradient-based, one-sided sampling method to reduce the number of training samples and a dedicated feature bundling method to bundle features. These two techniques improve model efficiency by reducing the number of samples and features, respectively. The unilateral sampling algorithm randomly retains all instances with large gradients and samples with small gradients, and the feature bundling algorithm bundles mutually exclusive features together, which can nondestructively reduce feature dimensionality.

(3) XGBoost is also based on the boosting ensemble strategy. XGBoost combines various weak learners by using additive training strategies and integration techniques to build powerful learners. To improve the computational accuracy, the loss function of the XGBoost model is extended using the second-order Taylor method. In addition, XGboost adds L1 and L2 regularization terms to avoid overfitting and reduce model complexity. The XGBoost algorithm also supports multiple threads for parallel computation, which speeds up the training and prediction of the model. XGBoost can also handle missing values automatically, eliminating the need for the manual processing of missing values. XGBoost algorithm also has a built-in cross-validation function, which can help users choose the best combination of parameters.

### 2.6. Classification Performances Evaluation

In this study, in order to more fully and accurately evaluate the models’ performances, five-fold cross-validation was applied on the EEG features to reduce bias in the models’ results, with 4 pieces of the samples used for training the model and 1 piece of the samples for testing the model to estimate the classification performances. The five-fold cross-validation method was repeated 10 times, and the final results were averaged. Here, the indicators of accuracy, precision, recall, and F1 score were computed for a comprehensive assessment of model performances. Furthermore, accuracy was defined as the proportion of correctly categorized samples to the total samples, precision as the proportion of correctly predicted positive categories to all the samples predicted as positive categories, recall as the proportion of correctly predicted positive categories to all the samples that are actually positive categories, and the F1 score as the reconciled mean of precision and recall. In order to calculate these four indicators, true positive (*TP*), false negative (*FN*), true negative (*TN*) and false positive (*FP*) are described as follows [45]: *TP* denotes the number of samples that are actually positive and correctly predicted as positive, *FP* denotes the number of samples that are actually negative but incorrectly predicted as positive, *FN* denotes the number of samples that are actually positive but incorrectly predicted as negative, and *TN* denotes the number of samples that are actually negative and correctly predicted as negative. Then, these four evaluation metrics for classification performances evaluation are defined as in Formulas (8)–(11).
(8)$$Accuracy=\frac{TP+TN}{TP+TN+FP+FN}$$
(9)$$Precision=\frac{TP}{TP+FP}$$
(10)$$Recall=\frac{TP}{TP+FN}$$
(11)$$F1=\frac{2TP}{2TP+FP+FN}$$

### 2.7. Statistical Analysis

One-way analysis of variance (ANOVA) was implemented to evaluate the statistical differences for the multi-dimensional EEG features between GAD and DD. Specifically, one-way ANOVA was carried out on the features of PSD, FE, and PLI. The threshold of *p* < 0.05 was considered a significant statistical difference.

## 3. Results

As shown in Figure 1, statistically significant differences in EEG power values between the two groups of patients were primarily observed in the alpha1 and beta rhythms. These differences were mainly distributed in the frontal, central, parietal, and temporal regions, while no statistically significant differences were found in the occipital region. Additionally, the power values in the DD group were significantly higher than those in the GAD group. As depicted in Figure 2, significant differences in FE were observed in the frontal region, specifically in the beta rhythm, while no significant differences were found in other rhythms. Furthermore, the fuzzy entropy in the DD group was higher than that in the GAD group.

In Figure 3, the brain network results for GAD and DD are depicted. As observed in the figure, abnormal functional connections were detected between different brain regions in both the GAD and DD groups across the four rhythms. These findings indicated alterations in the brain network structure. Specifically, in the theta, alpha1, and beta rhythms, higher functional connection weights were observed in the DD group compared to the GAD group, with ratios of 9/5, 10/5, and 23/8, respectively. In contrast, in the alpha2 rhythm, lower functional connection weights were found in the DD group compared to the GAD group, with a ratio of 4/8. The significance of dividing the alpha rhythm into alpha1 and alpha2 sub-rhythms in EEG-based brain fatigue research was indicated by the results. Klimesch has emphasized that using narrower frequency bands reduces the risk of frequency effects being canceled out or not discovered [46]. This study provides compelling evidence supporting the importance of such narrower frequency band divisions, as demonstrated by the results of the alpha1 and alpha2 rhythms. Additionally, the division of the frequency band into narrower sub-bands enhances the physiological meaning of these sub-bands and increases the statistical significance of their respective results.

Significant differences were observed in EEG features between the GAD and DD groups across three dimensions in the results presented in Figure 1, Figure 2 and Figure 3, providing a basis for the accurate diagnosis of GAD and DD. To address this further, three ensemble learning algorithms, namely RF, LightGBM, and XGBoost, were employed in the present study to identify significantly different EEG features and rhythms between GAD and DD, as summarized in Table 2 and Table 3. If these four evaluation metrics of accuracy, precision, recall, and F1 score are larger and closer together, it indicates better model performance. Table 2 demonstrated that the highest accuracy of 85.5 ± 0.3% was achieved by the PLI feature, followed by PSD and FE, with accuracies of 66.9 ± 0.3% and 82.2 ± 0.3%, respectively. These findings suggest that PLI more accurately reflects the brain functional differences between GAD and DD. Table 3 revealed that the combination of the three major feature categories from the four rhythms using the XGBoost classifier achieved the highest known classification accuracy of 99.1 ± 0.2%. Furthermore, the individual beta rhythm attained an accuracy of 98.3 ± 0.2%, which was significantly higher than that of the theta, alpha1, and alpha2 rhythms, and comparable to the accuracy obtained when all rhythms were combined. These results indicate that the beta rhythm is a more indicative factor of the brain functional differences between GAD and DD.

## 4. Discussion

In this study, we immersed ourselves in the exploration of EEG characteristics specific to GAD and DD. Our investigation encompassed an analysis of resting-state EEG data, considering a range of viewpoints, such as PSD, FE, and PLI. Our objective was to shed light on the underlying neurodynamic mechanisms that differentiate GAD from DD. Here are the key findings we uncovered. First and foremost, through a comprehensive analysis of PSD, FE, PLI, and classification methods, we observed significant differences in the beta rhythm between GAD and DD. These findings provide valuable insights into the distinctive features of these two disorders. Secondly, the highest accuracy among the three features was achieved by the PLI (85.5 ± 0.3%), which also provided the most accurate reflection of the functional brain imaging between GAD and DD. However, the overall feature accuracy reached its peak at 99.1 ± 0.2%, emphasizing the need for a comprehensive assessment rather than solely relying on the comparison of the PLI feature. A detailed analysis of the obtained results will be presented next.

The beta rhythm plays a crucial role in the study and clinical applications of mental disorders, as it provides valuable insights into the underlying neurophysiological mechanisms. Significant differences were found between GAD and DD, specifically in the beta rhythm, in this study. The importance of the beta rhythm will be further analyzed from various angles to shed light on its relevance in distinguishing between GAD and DD.

In comparison to GAD patients, higher PSD, FE, and functional connectivity weights in the beta rhythm are observed in DD patients across most brain regions. These differences are predominantly observed in the frontal, central, parietal, and temporal regions, while no statistical differences are found in the occipital region. The increased PSD values in the beta rhythm suggest a heightened state of brain activity and neural arousal, whereas the elevated FE values indicate increased complexity [11]. The findings of this study further highlight that DD is characterized by higher levels of activity and more complex prefrontal brain states compared with GAD. The beta rhythm is widely recognized to be associated with brain vigilance and wakefulness states [47]. Previous research has demonstrated significant differences in the functional connections in the beta frequency band between DD and healthy individuals [35]. The total number of key functional connections in the beta rhythm accounted for 43.1%, which was significantly higher than those found in the theta (19.4%), alpha1 (20.8%), and alpha2 (16.7%) rhythm. Moreover, the abnormal expression of the beta rhythm has garnered attention in some other EEG-based studies on mental diseases [23]. The high proportion of the beta rhythm within the all EEG characteristics would provide fundamental theoretical support for the determination of neurobiological markers in future DD and GAD diagnoses.

The changes in functional connectivity quantity represent variations in regional coordination and cognitive functioning in the brain and reflect the reorganization in GAD and DD [48,49,50]. Significant brain reorganization in multiple frequency ranges has been observed in DD [51]. Abnormal functional connectivity patterns have been reported to exhibit distinct characteristics corresponding to different symptoms of psychiatric disorders. For instance, reduced functional connections in the frontal lobe and other brain regions have been associated with GAD [23,52,53]. Enhanced connectivity in the frontal lobe region has been observed in patients with attention-deficit. In this study, the key functional connections were further applied to explore the mechanistic differences between GAD and DD. Significant reorganization was found in both disorders.

Machine learning has been widely applied in medical research to assist in the prediction and diagnosis of mental disorders by analyzing objective indicators of their mechanisms [54,55,56]. The highest classification accuracy of 98.3 ± 0.2% was gained by the beta rhythm, surpassing the theta (86.8 ± 0.7%), alpha1 (86.4 ± 0.9%), and alpha2 (90.5 ± 0.5%) rhythms. This signifies that the specificity of the beta rhythm between GAD and DD patients is further supported by the machine learning classification results. Considerable attention has been given to the beta rhythm and its relevance to GAD and DD in both academic research and clinical applications [35]. On the one hand, the EEG features of the beta rhythm have been quantified to assess GAD and DD, with positive correlations being found between the FE values in the beta rhythm. On the other hand, there is increasing evidence indicating a strong correlation between the EEG features of the beta rhythm and brain activity levels.

Studies have demonstrated that EEG features yield an optimal performance as classification features [35,57]. In this study, Multiple EEG features were extracted and classified with machine learning models (RF, LightGBM, and XGBoost) between GAD and DD, resulting in the highest classification accuracy of 99.1%, obtained using the XGBoost model. The highest accuracy of all rhythm features further confirmed the effectiveness of employing machine learning for the diagnosis of GAD and DD using EEG. Few studies have focused on classifying GAD and DD using machine learning models. These models in our study can better differentiate EEG features between GAD and DD. Among the three EEG features, the PLI (85.5 ± 0.3%) yielded the highest accuracy and provided the most accurate reflection of the functional brain imaging between GAD and DD. Similar results were found by Li et al. [35]. However, the overall feature accuracy peaked at 99.1 ± 0.2, highlighting the importance of considering a comprehensive approach rather than solely comparing the PLI feature. To date, no studies have classified the EEG features of GAD and DD using machine learning models. This study successfully achieved a high level of accuracy in classifying individuals with GAD and DD. The findings present compelling and objective scientific evidence supporting the clinical categorization of patients with GAD and DD.

There are a few limitations in this manuscript that should be acknowledged. Firstly, the sample size in the study consisted of 38 individuals with GAD and 34 volunteers with DD, which may be considered relatively small for making definitive conclusions. Secondly, the study utilized an EEG system with only 16 electrodes, and it would be beneficial to carry out future research using high-density EEG to validate the current findings.

## 5. Conclusions

This study took an innovative approach to delve into the functional brain imaging of DD and GAD by examining the similarities and differences between multi-dimensional EEG features via one-way ANOVA and machine learning techniques. Notably, the analysis revealed noteworthy distinctions within the PLI features and beta rhythm characteristics. This unearths compelling neurophysiological evidence that has potential diagnostic implications for both DD and GAD. As there is a great deal of similarity in the clinical diagnosis of DD and GAD, such as similarities in symptoms and very close scale scores, the findings from this study contribute to the classification, diagnosis, and treatment of individuals with DD and GAD in clinical practice. In future studies, we hope to further clarify the EEG eigenvalues with significant variability for visualization and quantitative diagnosis, and to develop accurate, stable and reliable diagnostic models for clinical use.

## Figures and Tables

**Figure 1 brainsci-13-01282-f001:**
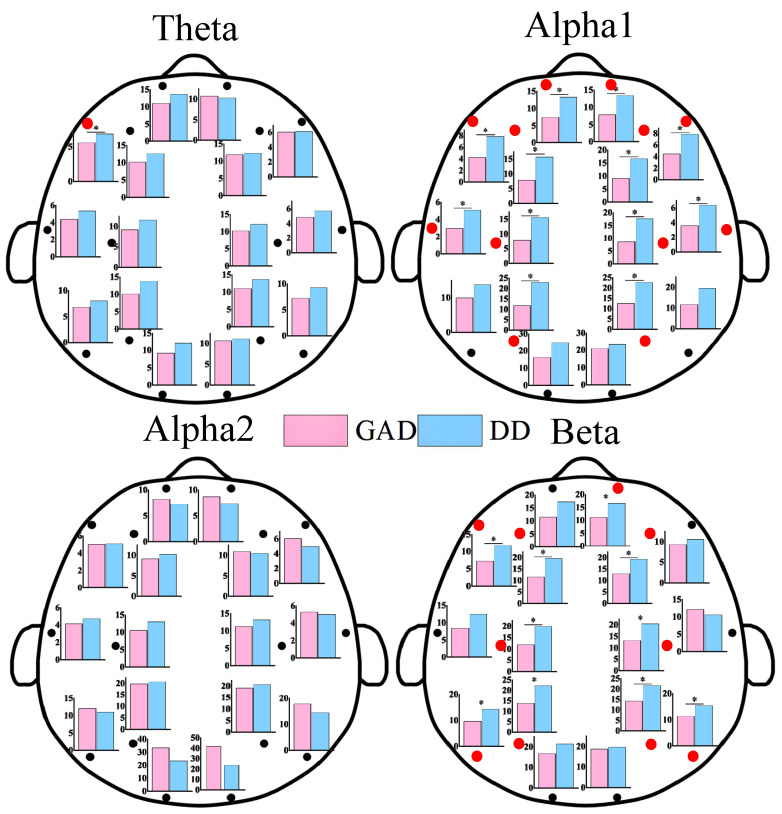
EEG power results of each EEG channel between GAD and DD for four EEG rhythms. * means that the power of the corresponding EEG electrode positions has significant statistical differences (*p* < 0.05). We also use red dots to highlight locations with statistically significant differences.

**Figure 2 brainsci-13-01282-f002:**
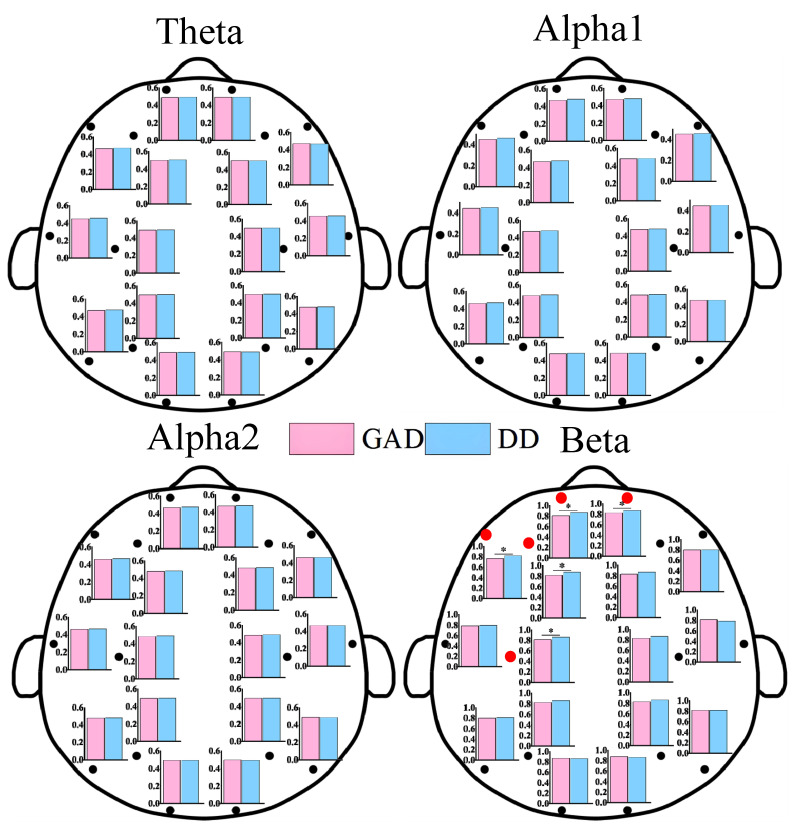
FE results of each EEG channel between GAD and DD for four EEG rhythms. * means that the FE of the corresponding EEG electrode positions has significant statistical differences (*p* < 0.05). We also use the red dots to highlight locations that have statistically significant differences.

**Figure 3 brainsci-13-01282-f003:**
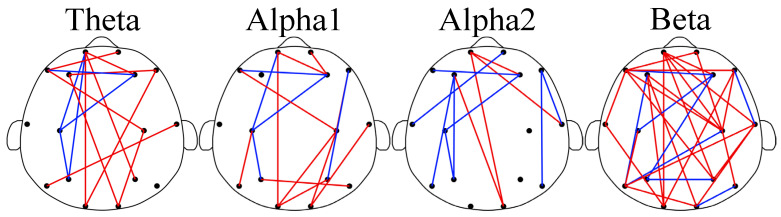
Brain networks of the four rhythms. In the subgraph, the red edge represents the lower PLI strength of GAD compared with DD; the blue edge represents the opposite.

**Table 1 brainsci-13-01282-t001:** Demographic and clinical characteristics of the participants.

Characteristics	GAD (*n* = 38)	DD (*n* = 34)	*p*-Value
Age (year)	41.2 ± 7.2	42.3 ± 9.6	>0.05
Gender: male/female	10/28	10/24	—
HAMA	26.2 ± 5.8	—	—
HAMD-17	—	26.3 ± 4.9	—

**Table 2 brainsci-13-01282-t002:** Classification performances of different feature types.

Feature	Models	Accuracy (%)	Precision (%)	Recall (%)	F1 (%)
PSD	RF	66.0 ± 0.4	69.3 ± 0.8	42.0 ± 0.5	52.3 ± 0.4
	LightGBM	66.0 ± 0.4	66.8 ± 0.6	46.3 ± 0.6	54.7 ± 0.4
	XGboost	66.9 ± 0.3	64.9 ± 0.6	55.4 ± 0.5	59.8 ± 0.4
FE	RF	76.9 ± 0.4	78.8 ± 0.5	65.5 ± 0.3	71.5 ± 0.3
	LightGBM	76.0 ± 0.4	75.9 ± 0.5	67.5 ± 0.5	71.5 ± 0.4
	XGboost	82.2 ± 0.3	81.3 ± 0.2	77.9 ± 0.3	79.6 ± 0.2
PLI	RF	78.3 ± 0.4	80.8 ± 0.4	67.1 ± 1.0	73.3 ± 0.5
	LightGBM	79.8 ± 0.3	79.4 ± 0.3	73.6 ± 0.6	76.4 ± 0.3
	XGboost	**85.5 ± 0.3**	**85.1 ± 0.5**	**81.6 ± 0.6**	**83.3 ± 0.3**

**Table 3 brainsci-13-01282-t003:** Classification performances of different rhythms.

Rhythm	Models	Accuracy (%)	Precision (%)	Recall (%)	F1 (%)
Theta	RF	79.5 ± 0.7	83.4 ± 0.5	67.3 ± 1.5	74.5 ± 1.0
	LightGBM	82.6 ± 0.8	82.5 ± 0.5	77.3 ± 1.2	79.8 ± 0.9
	XGboost	86.8 ± 0.7	86.7 ± 0.5	82.9 ± 1.4	84.7 ± 0.9
Alpha1	RF	79.3 ± 0.5	87.95 ± 1.2	62.0 ± 0.6	72.7 ± 0.7
	LightGBM	82.6 ± 0.6	83.8 ± 0.7	75.3 ± 0.8	79.3 ± 0.6
	XGboost	86.4 ± 0.9	86.7 ± 0.8	81.9 ± 1.3	84.2 ± 1.0
Alpha1	RF	85.1 ± 0.6	87.2 ± 0.7	77.9 ± 0.9	82.3 ± 0.8
	LightGBM	87.2 ± 0.6	86.3 ± 0.4	84.5 ± 1.3	85.4 ± 0.8
	XGboost	90.5 ± 0.5	90.0 ± 0.6	88.5 ± 1.1	89.2 ± 0.6
Beta	RF	94.6 ± 0.3	95.5 ± 0.4	92.2 ± 0.7	93.8 ± 0.4
	LightGBM	97.0 ± 0.3	96.8 ± 0.4	96.5 ± 0.4	96.7 ± 0.3
	XGboost	98.3 ± 0.2	98.1 ± 0.4	98.1 ± 0.3	98.1 ± 0.3
All rhythms	RF	95.6 ± 0.3	98.1 ± 0.4	91.9 ± 0.7	94.9 ± 0.3
	LightGBM	98.2 ± 0.3	98.2 ± 0.3	97.9 ± 0.6	98.0 ± 0.3
	XGboost	**99.1 ± 0.2**	**99.1 ± 0.1**	**98.8 ± 0.5**	**99.0 ± 0.3**

## Data Availability

Not applicable.

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
