# Peer review of "Neuroimaging Study of Brain Functional Differences in Generalized Anxiety Disorder and Depressive Disorder"

_brainsci, 2023, doi:10.3390/brainsci13091282_

Round 1

Reviewer 1 Report

The article entitled “Brain Functional Differences between Generalized Anxiety Disorder and Depression Disorder” by Xuchen Qi and Wanxiu Xu submitted to brain sciences MDPI consists in a study on a sample of 38 patients with Generalized Anxiety Disorder (GAD) and 34 patients with Depressive Disorder (DD) aged between 18 and 55 years enrolled from the psychiatry department aims to study the neuro-mechanism differences between GAD and DD by multidimensional electroencephalogram (EEG) characteristics detection assisted with machine learning.

Starting from the consideration that GAD and DD are characterized by distinct disorders with unique underlying neurobiological mechanisms, the research aims to study the neuro-mechanism differences between GAD and DD through the use of multidimensional electroencephalogram (EEG) characteristics detection assisted with machine learning. The scope of the study is to develop an enhanced screening framework that can effectively differentiate between GAD and DD as potential biomarker to contribute to proper diagnosis, personalized treatment and improved outcome.

A total of 10-minute resting-state EEG signals were recorded from 38 GAD and 34 DD individuals. Multidimensional features of the EEG were extracted and analysed.

Participants were diagnosed using the Structural Interview based on Diagnostical and Statistical Manual of Mental Disorder, 5th Edition (DSM-5) criteria. Each subject completed the Hamilton Anxiety Rating Scale (HAM-A) and the 17-item Hamilton Depression Rating Scale (HAMD-17).

The findings highlight that DD has significantly higher PSD values in the alpha1 and beta rhythms compared to GAD, and the frontal region of DD showed significantly higher FE in the beta rhythm compared to GAD. Moreover, aberrant functional connectivity between brain regions was observed in all four rhythms between GAD and DD, indicating significant differences in brain network structures.

The paper is certainly innovative and interesting. The introduction is well structured and well written. Methods are well designed and purposes of the study are clear and well-defined. The division in specific paragraphs is adequate and useful for the readers.

Authors should definitely pay attention to the use of the specific terms. In the first paragraph of Materials and Methods section, the Diagnostic and Statistical Manual of Mental Disorder, Fifth Edition, is abbreviated as DSM-V, but it correctly named as DSM-5 in professional literature. The same, HAMD-17 stands for Hamilton Depression Rating Scale (HAMD-17), not for “Hamilton Rating Scale for Depression”.

May it be possible for the authors to provide a table in which the socio-demographic and clinical features of the participants are summarized? Even though the clinical variables of the sample are not the main focus of the study, I think that could be interesting to add them in the text or as supplementary material.

The discussion of the finding is interesting and food for thoughts. Limitations of the study are correctly specified. Conclusions summarized the elements of novelty of the research.

Reviewer 2 Report

The overall manuscript is written nicely and it’s clear to understand. A few of my minor comments are given below.

Line 93: How authors defined these patients with GAD and DD, it would be better to give basic criteria over this selection.

Figure 1,2: Define the n and significance value in the figure legend

Do these patients have any history of behavior impairments and does it affect the Theta, Alpha1, Alpha 2, and Beta rhythms? If so, that might influence the outcome of the study. 

The English language is fine. 

Reviewer 3 Report

27 July 2023 

The review report on the manuscript, titled “Brain Functional Differences between Generalized Anxiety Disorder and Depression Disorder” by Qi and Xu, submitted to Brain Sciences.

Manuscript ID: brainsci-2547070 

Dear Authors, 

The current challenge in understanding the distinctions between Generalized Anxiety Disorder (GAD) and Depressive Disorder (DD) involves finding objective biomarkers. Traditionally, the diagnosis and differentiation of these mental health conditions have relied heavily on subjective assessments and symptom-based criteria, leading to potential misdiagnoses and delayed personalized treatment. Advancements in machine learning and neuroimaging technologies have shown promise in analyzing electroencephalography (EEG) data to identify brain functional differences associated with GAD and DD. However, the field still faces the challenge of effectively integrating these technologies and developing robust classification models that can accurately classify individuals with GAD and DD based on EEG features. In the present research article titled “Brain Functional Differences between Generalized Anxiety Disorder and Depression Disorder,” Qi and Xu aim to identify specific brain functional differences between GAD and DD, which can contribute to accurate diagnosis and personalized treatment. The introduction highlights the impact of mental health conditions on well-being and the need for objective biomarkers to distinguish between GAD and DD. The paper discusses advancements in machine learning and neuroimaging technologies as potential tools for this purpose. Results reveal significant differences in EEG power values, FE, and functional connectivity weights in the beta rhythm between GAD and DD patients. The beta rhythm demonstrates the highest classification accuracy in distinguishing between the two disorders. The discussion emphasizes the importance of the beta rhythm in distinguishing GAD from DD and its relevance to brain activity levels. The machine learning models achieve high accuracy in classifying EEG features between GAD and DD, with PLI providing the most accurate reflection of brain functional differences. In summary, the manuscript presents a comprehensive study utilizing machine learning and EEG data to differentiate between GAD and DD. The results highlight the significance of the beta rhythm and demonstrate promising potential for accurate diagnosis and personalized treatment strategies for individuals with GAD and DD.

The primary strength of this manuscript lies in its comprehensive approach to investigating and differentiating between GAD and DD using (EEG) data and machine learning techniques. In general, I think the idea of this article is really interesting, and the authors’ fascinating observations on this timely topic may be of interest to the readers of Brain Sciences. However, some comments, as well as some crucial evidence that should be included to support the author’s argumentation, needed to be addressed to improve the quality of the manuscript, its adequacy, and its readability prior to its publication in the present form. My overall opinion is to publish this research article after the author has carefully considered my comments and suggestions below.

Please consider the following comments:

1.      First, please expand all abbreviations in their initial occurrence and avoid using abbreviations in subheadings.

2.      I recommend revising the title. In its current form, I find it to be relatively long and can be streamlined without losing essential information. In my opinion, the study could benefit from a more concise and clear title that captures the core focus of the research. Possible suggestion: "Neuroimaging Study of Brain Functional Differences in Generalized Anxiety Disorder and Major Depressive Disorder" [1–3].

3.      A graphical abstract that will visually summarize the main findings of the manuscript is highly recommended.

4.      Abstract: I would like the authors to make as much effort for this section as for the rest of the manuscript. Overall, the abstract provides a concise overview of the study and its key findings. However, there are some areas that could be improved or clarified to enhance its clarity and effectiveness, according to the guidelines of the Journal [4]. First of all, this section could benefit from breaking some of the longer sentences into smaller ones. For example, the first sentence can be split into two to make it easier to read and understand. Furthermore, the abstract mentions "multidimensional electroencephalogram (EEG) characteristic detection assisted with machine learning," but it doesn't specify the exact EEG characteristics or the machine learning algorithms used. I would ask the authors to provide more details, which would make the abstract more informative. That having said, I would like the authors to reorganize the main subsections with 200–220 words, max. 250 words, proportionally presenting the background, methods, results, and conclusion. The background should include the general background (one to two sentences), the specific background (two to three sentences), and "the current issue addressed to this study" (one sentence), leading to the objectives. In this subsection, I would like the authors to lay out basic information, a problem statement, and their motivation to break off. The methods should clarify the authors’ approach, such as study design and variables, to solving the problem and/or making progress on the problem. The results should close with a single sentence putting the results in a more general context. The conclusion should open with one sentence describing the main result using such words as “Here we show”, which should be followed by statements such as the potential and the advance this study has provided in the field, and finally a broader perspective (two to three sentences) readily comprehensible to a scientist in any discipline [5–8].

5.      Keywords: Please list ten keywords chosen from Medical Subject Headings (MeSH) [9] and use as many as possible in the title and in the first two sentences of the abstract [7,8]. I would suggest “Neurobiology” as a keyword. Please choose the keyword listed in MeSH for “depression disorder” such as “depression” or “major depressive disorder (MDD)”

6.      Introduction: The authors need to reorganize this section with several paragraphs totaling about 1000 words, introducing information on the key study constructs that should be understandable to readers from any discipline, and making it persuasive enough to advance the main purpose of the recent research the author has conducted and the particular purpose the author has intended by this protocol. I would like to encourage the authors to start the introduction with a general background, then move on to the specific background on established neural substrates and brain regions commonly implicated in GAD and DD by mentioning key studies and findings that have identified specific brain regions, such as the amygdala, prefrontal cortex, and hippocampus, as key players in these disorders, and finally the current issue addressed to this study, leading to the objectives. Those main structures should be organized in a logical and cohesive manner [10].

7.      In this regard, I believe that the Introduction section would benefit from additional information to enhance its clarity and contextualization. To strengthen this section, I suggest highlighting the importance of EEG as a non-invasive technique for studying brain function and how it can offer valuable insights into the dynamic neural activity associated with GAD and DD. Please discuss the previous research that has shown EEG to be sensitive to alterations in brain activity and connectivity in psychiatric disorders, emphasizing the potential of EEG to uncover distinct neural substrates in GAD and DD [11–12]. Furthermore, acknowledge any gaps or limitations in our current understanding of the neural substrates in GAD and DD. This may include discrepancies in findings, variations in brain network connectivity, or a lack of precise biomarkers that can reliably differentiate the two disorders. Emphasize that the current study aims to address these gaps by utilizing multi-dimensional EEG features and ensemble learning models [13]. Explicitly state the significance of using ensemble learning methods to identify and classify EEG features associated with GAD and DD. Discuss how ensemble learning can harness the strengths of multiple machine learning algorithms to improve accuracy and overcome potential biases that might arise from individual algorithms. By adding these extra points to the introduction, the manuscript will give a more complete overview of the neural bases of GAD and DD, as well as the possible roles of EEG-based analysis and ensemble learning. This will undoubtedly enhance the significance and impact of the research.

8.      Methods (Multidimensional EEG Features Computation): I recommend opening this section with a short introductory paragraph regarding the study design and methodology and citing more references to ensure the reliability and integrity of the evidence in the study design the authors built and the methodology they have decided to apply. Consider providing a brief explanation of what "bagging" and "boosting" mean in the context of ensemble learning to make it easier for readers who may not be familiar with these concepts to understand the methods used. Also, I would suggest providing a short description of each learning models (RF, LightGBM, and XGBoost) short description, strengths and weaknesses. This will help readers understand why these particular models were chosen for the study.

9.      Methods (Classification Performances Evaluation): I would ask the authors to clarify the reason for choosing 5-fold cross-validation and repeating it 10 times and to add a clear definition of the evaluation metrics used, such as accuracy, precision, recall, and F1 score. It is important to explain these metrics to readers who may not be familiar with them.

10.  Results: In the results section, it would be beneficial to present the data in a more organized manner, such as by using tables or graphs in different colors, as they are presented mainly in a descriptive way. This will make it easier for readers to interpret the results and fully discuss the observed differences between GAD and DD. Also, in my opinion, authors should consider adding more quantitative analysis, such as p-values or effect sizes, to support the claims made in the text. Additionally, the classification results (Tables 1 and 2) are impressive but require more context. Please explain the meaning of each classification accuracy and consider providing information about sensitivity, specificity, and other relevant metrics to offer a more comprehensive assessment of the models' performance.

11.  Discussion: The discussion section lacks a clear and structured organization. I would like the authors to begin this section with an introduction and then provide a summary of the previous section. Then, I expect the authors to develop arguments clarifying the potential of this study as an extension of the previous work, the implication of the findings, how this study could facilitate future research, the ultimate goal, the challenge, the knowledge and technology necessary to achieve this goal, the statement about this field in general, and finally the importance of this line of research. It is particularly important to present its limits, its merits, and the potential translation of this protocol into clinical practice. Particularly, here the authors should provide more in-depth analysis and interpretation of the results for example, discussing why the beta rhythm shows pronounced differences between GAD and DD, and how this finding aligns with existing research on these disorders [14,15]. I suggest closing this section with a paragraph that puts the results into a more general context.

12.  Conclusion: I believe that presenting this section with 150–200 words would benefit from a single paragraph that presents some thoughtful and in-depth considerations by the authors as experts in order to convey the main message. The authors should make an effort to explain the theoretical implications as well as the translational application of their research. In order to understand the significance of this field, I believe it would be necessary to discuss theoretical and methodological avenues in need of refinement as well as future research directions.

13.  References: Please revise the bibliography according to the journal guidelines [16], as there are several incorrect citations. Indeed, according to the Journal’s guidelines, the authors should provide the abbreviated journal name in italics and punctuated with periods. Also, please cite more references. An original article like this typically cites over 6070 references.

Overall, the manuscript contains two figures, two tables, and 46 references. I believe that this manuscript contributes to the growing body of knowledge regarding the neural distinctions between GAD and DD. The combination of EEG characteristics and machine learning techniques holds promise for advancing psychiatric diagnostics and improving patient outcomes by providing a deeper understanding of the underlying neurobiological mechanisms of these disorders. I hope that, after careful revisions, the manuscript can meet the journal’s high standards for publication.

Best regards, 

Reviewer 

References:

  1. https://plos.org/resource/how-to-write-a-great-title/
  2. https://www.nature.com/nature-index/news-blog/how-to-write-a-good-research-science-academic-paper-title
  3. https://www.indeed.com/career-advice/career-development/catchy-title
  4. https://www.mdpi.com/journal/brainsci/instructions
  5. https://www.scribbr.com/dissertation/abstract/
  6. https://writing.wisc.edu/handbook/assignments/writing-an-abstract-for-your-research-paper/
  7. https://doi.org/10.5812/ijem.100159
  8. https://doi.org/10.4103/sja.SJA_685_18
  9. https://meshb.nlm.nih.gov/
  10. https://dept.writing.wisc.edu/wac/writing-an-introduction-for-a-scientific-paper/
  11. https://doi.org/10.17219/acem/165944
  12. https://doi.org/10.1016/j.neubiorev.2023.105163
  13. https://doi.org/10.17219/acem/166476
  14. https://doi.org/10.3163/1536-5050.103.2.001
  15. https://www.scribbr.com/dissertation/discussion/
  16. https://www.mdpi.com/journal/brainsci/instructions 

27 July 2023 

The review report on the manuscript, titled “Brain Functional Differences between Generalized Anxiety Disorder and Depression Disorder” by Qi and Xu, submitted to Brain Sciences.

Manuscript ID: brainsci-2547070 

Dear Authors,

Based on the English proficiency assessment, it is noted that minor editing of the English language is required. While the overall communication is clear and understandable, there are some areas that could benefit from slight improvements in grammar, syntax, and word choice. Attention to detail, such as refining sentence structure and ensuring proper tense usage, will enhance the overall coherence and fluency of the written work. With some minor editing adjustments, the English language proficiency can be further enhanced.

Best regards,

Reviewer

Round 2

Reviewer 3 Report

21 August 2023 

The 2nd review report on the manuscript, titled “Brain Functional Differences between Generalized Anxiety Disorder and Depression Disorder” by Qi and Xu, submitted to Brain Sciences.

Manuscript ID: brainsci-2547070 

Dear Authors, 

In the present research article titled “Brain Functional Differences between Generalized Anxiety Disorder and Depression Disorder,” Qi and Xu aim to identify specific brain functional differences between generalized anxiety disorder (GAD) and depressive disorder (DD), which can contribute to accurate diagnosis and personalized treatment.I am pleased to see that the authors have addressed the issues raised in the previous round of the peer review session. Currently, the manuscript is a well-written research article with an informative layout that presents a comprehensive approach to investigating and differentiating between GAD and DD using (EEG) data and machine learning techniques. I am confident that the paper is of sufficient quality to warrant publication in the journal. Here, I leave a couple of suggestions to further improve the quality of the manuscript and thus ensure the readability of this article. I hope that the same authors will continue to publish more work.

1.      Abstract: I would like the authors to reorganize the main subsections with “200–220 words, max. 250 words”, proportionally presenting the background, methods, results, and conclusion.

2.      Discussion: I recommend not using subheadings in this section.

Best regards, 

Reviewer

21 August 2023 

The review report on the manuscript, titled “Brain Functional Differences between Generalized Anxiety Disorder and Depression Disorder” by Qi and Xu, submitted to Brain Sciences.

Manuscript ID: brainsci-2547070 

Dear Authors,

After evaluating the English proficiency, it has been determined that some minor revisions to the English language are necessary. While the overall communication is clear and understandable, certain areas could benefit from slight improvements in grammar, syntax, and word choice. Paying attention to detail, such as refining sentence structure and ensuring proper tense usage, will enhance the coherence and fluency of the written work as a whole. Making minor editing adjustments can lead to an improvement in English language proficiency.

Best regards,

Reviewer
